# Heterozygosity at a conserved candidate sex determination locus is associated with female development in the clonal raider ant (*Ooceraea biroi*)

Kip D Lacy[1]*, Jina Lee[1], Kathryn Rozen-Gagnon[2,3], Wei Wang[3], Thomas S Carroll[3], Daniel JC Kronauer[1,4]*

[1]Laboratory of Social Evolution and Behavior, The Rockefeller University, New York, United States; [2]Department of Molecular Genetics, University of Toronto, Toronto, Canada; [3]Bioinformatics Resouce Center, The Rockefeller University, New York, United States; [4]Howard Hughes Medical Institute, Chevy Chase, United States

## eLife Assessment

This study provides **valuable** insights into the evolutionary conservation of sex determination mechanisms in ants by identifying a candidate sex-determining region in a parthenogenetic species. It uses **solid**, well-executed genomic analyses based on differences in heterozygosity between females and diploid males. While the candidate locus awaits functional validation in this species, the study provides **convincing** support for the ancient origin of a non-coding locus implicated in sex determination.

**\*For correspondence:**
kipdlacy@gmail.com (KDL);
dkronauer@rockefeller.edu
(DJCK)

**Competing interest:** The authors declare that no competing interests exist.

**Abstract** Sex determination is a developmental switch that triggers sex-specific developmental programs. This switch is 'flipped' by the expression of genes that promote male- or female-specific development. Many lineages have evolved sex chromosomes that act as primary signals for sex determination. However, haplodiploidy (males are haploid and females are diploid), which occurs in ca. 12% of animal species, is incompatible with sex chromosomes. Haplodiploid taxa must, therefore, rely on other strategies for sex determination. One mechanism, 'complementary sex determination' (CSD), uses heterozygosity as a proxy for diploidy. In CSD, heterozygosity at a sex determination locus triggers female development, while hemizygosity or homozygosity permits male development. CSD loci have been mapped in honeybees and two ant species, but we know little about their evolutionary history. Here, we investigate sex determination in the clonal raider ant, *Ooceraea biroi*. We identified a 46 kb candidate CSD locus at which all females are heterozygous, but most diploid males are homozygous for either allele. As expected for CSD loci, the candidate locus has more alleles than most other loci, resulting in a peak of nucleotide diversity. This peak negligibly affects the amino acid sequences of protein-coding genes, suggesting that heterozygosity of a non-coding genomic sequence triggers female development. This locus is distinct from the CSD locus in honeybees but homologous to a CSD locus mapped in two distantly related ant species, implying that this molecular mechanism has been conserved since a common ancestor that lived approximately 112 million years ago.

## Introduction

Whether an animal develops as a male or a female is typically determined by a switch early in development. This switch, known as sex determination, is a multistep process beginning with primary sex determination signals triggering sex-specific splicing of downstream transcription factors that

**eLife digest** In most animals, sex is determined at conception. The factors that determine whether an embryo becomes male or female vary across animals, one example being sex chromosomes. In humans and other mammals, a gene on the Y chromosome triggers male development, whereas its absence allows female development.

A different system occurs in haplodiploid species, including wasps, bees and ants. In these animals, haploid eggs (which inherit one set of each chromosome from their mother) develop into males, while diploid eggs (which inherit two sets of chromosomes, one from each parent) develop into females. In many haplodiploid species, inbreeding can produce diploid males, leading to the idea of a system called complementary sex determination, in which having two different versions of a certain gene produces a female while having two identical versions (homozygous diploids) or just one version produces a male.

So far, genes or candidate genes responsible for this type of development have only been identified in honeybees and two ant species. Lacy et al. studied the clonal raider ant, *Ooceraea biroi*, which is evolutionarily distinct from those ants. The researchers sequenced the genomes of diploid males and females to look for regions of chromosomes where males had two identical alleles, but females had two different ones. They found such a region, a nearly 50,000 DNA base pair long stretch on chromosome 4 that was evolutionarily related to a non-coding RNA region found in the other two ant species.

In this region, they also found patterns of genetic diversity consistent with a complementary sex determination locus. These patterns were not found near genes related to the gene *transformer*, which are complementary sex determination loci in honeybees and possibly one of the ant species. This suggests that the long non-coding RNA–based locus may have been conserved for over 100 million years of ant evolution, while the *transformer*-derived system may have evolved later in some species, possibly independently adopting a similar function to that in honeybees.

Understanding complementary sex determination is crucial for the conservation of species, especially for pollinators. When populations shrink and inbreeding occurs, sterile diploid males can be produced, threatening population stability. Knowing which species use complementary sex determination can help track inbreeding and guide breeding programs. The discovery that a single complementary sex determination locus appears to be conserved across ants suggests that similar conservation strategies could apply to many species.

encode sexual identity (*Williams and Carroll, 2009*). Although the downstream transcription factors are evolutionarily conserved, primary sex determination signals are evolutionarily labile and comprise a diverse range of environmental and genetic mechanisms (*Bachtrog et al., 2014*). Genetically encoded primary signals (sex-determining genes or chromosomes) are usually found in only one sex. Such systems do not work in haplodiploid taxa (an estimated 12% of animal species, including ants; *Normark, 2003*), where any allele on any homologous chromosome can be transmitted from diploid females to their haploid male sons, and all alleles experience selection to be viably transmitted from both sexes (*Whiting, 1935*). Haplodiploids, therefore, require alternative sex determination mechanisms, but we know little about the molecular details.

Some parasitoid wasps use a mechanism involving maternal imprinting (*Verhulst et al., 2010*; *Zou et al., 2020*). However, in many Hymenoptera, inbred crosses produce diploid males, which led to the hypothesis that female development is triggered by heterozygosity as a proxy for diploidy (*Whiting, 1933*; *Whiting, 1943*). Under this 'complementary sex determination' (CSD) hypothesis, different alleles at a given sex determination locus 'complement' one another, meaning that heterozygosity triggers female development (*Figure 1a*). By contrast, the presence of a single allele (either because the individual is haploid or because a diploid individual is homozygous at the sex determination locus) permits male development. Under CSD, diploid males produce diploid sperm and, therefore, are functionally sterile (*van Wilgenburg et al., 2006*). This has several evolutionary consequences (*Cook and Crozier, 1995*); at the population level, there is selection against homozygosity, often meaning obligatory avoidance of inbreeding (*Page, 1980*). At the gene level, there is negative frequency-dependent

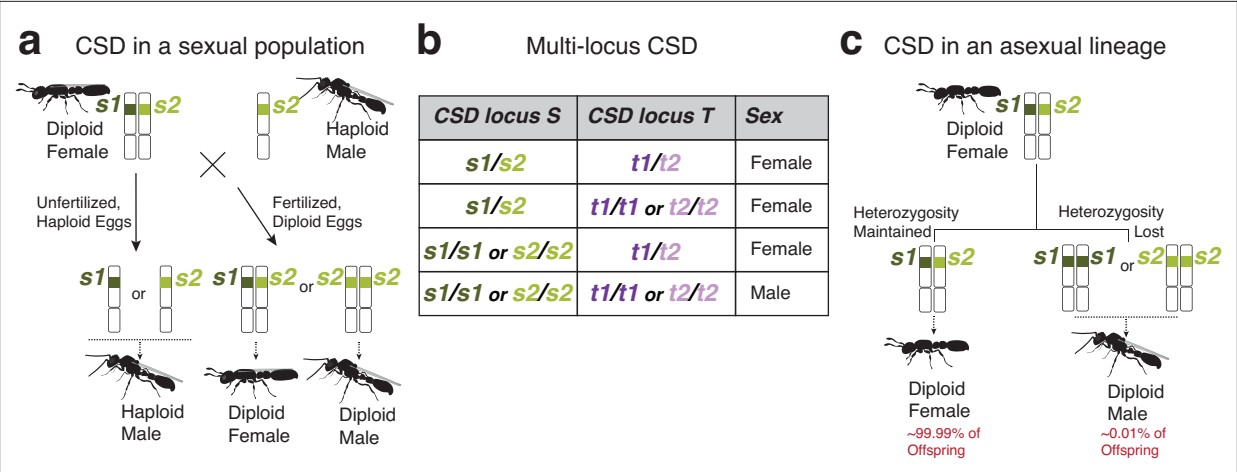

**Figure 1.** Theoretical expectations for complementary sex determination (CSD). (**a**) Cartoon depicting a mating between a diploid female with two different alleles at a sex determination locus and a haploid male bearing one of these two alleles. Half of the sexually produced diploid offspring from this mating are expected to develop as males. If many alleles co-occur in sexual populations, such matings will be rare. (**b**) Table illustrating multi-locus CSD in diploids, in which heterozygosity of at least one sex determination locus is required for female development. By contrast, only homozygosity at all loci results in diploid male development. (**c**) Cartoon depicting how CSD might work in asexual species such as the clonal raider ant. If diploid males arise from losses of heterozygosity at sex loci, then homozygosity for either of the two alleles should trigger male development. Offspring proportions reflect empirical observations in the clonal raider ant (*Kronauer et al., 2012*; *Oxley et al., 2014*).

(balancing) selection on CSD loci, resulting in the co-occurrence of many alleles at similar frequencies within populations (*Laidlaw et al., 1956*).

Having been a favored hypothesis for more than half a century, CSD was demonstrated in the honeybee *Apis mellifera* in 2003 (*Beye et al., 2003*). In *A. mellifera*, biallelic heteromers of the protein encoded by the *complementary sex determiner* (*csd*) gene trigger female development, whereas monoallelic homomers permit male development (*Beye et al., 2003*; *Beye et al., 2013*; *Otte et al., 2023*; *Seiler and Beye, 2024*). Although this primary sex determination signal differs from primary signals in other insects, it nonetheless converges upon the pathway that transduces sex determination signals in holometabolous insects (*Bopp et al., 2014*; *Sawanth et al., 2016*; *Wexler et al., 2019*). The Csd protein sex-specifically splices the closely linked splicing factor *feminizer* (*fem*) (the honeybee version of *transformer* [*tra*] in *Drosophila* and other insects), which in turn leads to the production of sex-specific isoforms of *doublesex* (*dsx*), a transcription factor that regulates sex-specific development (*Hasselmann et al., 2008*; *Gempe et al., 2009*; *Verhulst et al., 2010*; *Zou et al., 2020*). Intriguingly, *csd* is a paralog of *fem*. Following the prediction that CSD loci should evolve under balancing selection, many *csd* alleles are held at similar frequencies, and the locus bears signatures of adaptive evolution (*Hasselmann and Beye, 2004*), with amino acid-level heterozygosity in certain protein domains required for female development (*Otte et al., 2023*).

CSD is assumed to occur in many Hymenoptera that occasionally produce diploid males (*van Wilgenburg et al., 2006*). Until recently, however, heterozygosity-dependent female development had not been demonstrated outside of *A. mellifera*, which remained the only species for which a CSD locus had been genetically mapped. A recent genetic mapping study in the wasp *Lysiphlebus fabarum* identified at least one and as many as four candidate sex determination loci (*Matthey-Doret et al., 2019*) (the multi-locus extension of the CSD hypothesis proposes that heterozygosity at any of multiple sex determination loci is sufficient to trigger female development [*Crozier, 1971*; *Figure 1b*]). However, a requirement of heterozygosity at these loci for female development was not demonstrated. In the ant *Vollenhovia emeryi*, a mapping study found two sex determination QTL (*Miyakawa and Mikheyev, 2015*), with follow-up work suggesting that these function as a multi-locus CSD system (*Miyakawa and Miyakawa, 2023*). One of these QTL (*V.emeryiCsdQTL1*) contains two closely linked *tra* homologs, reminiscent of the locus containing *csd* and *fem* in *A. mellifera*. By contrast, the other QTL (*V.emeryiCsdQTL2*) only contains genes without annotated functions in sexual development. Although *V. emeryi* sex determination is likely integrated through *tra* and *dsx*, the

molecular basis of their primary sex determination signals remains unclear (*Miyakawa et al., 2018*; *Miyakawa and Miyakawa, 2023*).

Recently, a sex determination locus was identified in the Argentine ant, *Linepithema humile* (*Pan et al., 2024*). Consistent with CSD, all pairwise heterozygous combinations of the seven identified alleles were found in females, whereas diploid males were invariably homozygous for one of the seven alleles. Moreover, these alleles occurred at roughly equal frequencies within populations, suggesting evolution under balancing selection. The mapped locus is located within a noncoding genomic region tightly linked to *ant noncoding transformer splicing regulator* (*ANTSR*), a long noncoding RNA (lncRNA) with previously uncharacterized function. *ANTSR* knockdown in female-destined embryos leads to male-specific splicing of *tra*, suggesting that this lncRNA transduces the primary sex determination signal encoded by the heterozygosity (or hemi- or homozygosity) of the linked noncoding region (*Pan et al., 2024*). Because the synteny of this region (including the presence of a lncRNA putatively homologous to *ANTSR*) is conserved across the ants, bees, and vespoid wasps, it was suggested that this sex determination locus may be deeply conserved among the Aculeata (*Pan et al., 2024*). However, synteny is insufficient to demonstrate functional conservation, and genetic mapping of sex determination has yet to be performed in most of these taxa.

To explore the evolution of sex determination across ants, we investigated the topic in the clonal raider ant, *Ooceraea biroi*. Diploid males occur sporadically in this species, suggesting that, like other ants, they might employ CSD (*Kronauer et al., 2012*). *O. biroi* reproduces via a mode of parthenogenesis in which two haploid nuclei from a single meiosis fuse to produce diploid offspring (central fusion automixis) (*Oxley et al., 2014*). Although heterozygosity is maintained with high fidelity despite meiotic crossover recombination (*Lacy et al., 2024*), losses of heterozygosity occur occasionally, probably due to the inheritance of one recombined and one non-recombined version of a homologous chromosome (*Kronauer et al., 2012*; *Oxley et al., 2014*; *Trible et al., 2023*; *Lacy et al., 2024*). Therefore, if *O. biroi* uses CSD, diploid males might result from losses of heterozygosity at sex determination loci (*Figure 1c*), similar to what is thought to occur in other asexual Hymenoptera that produce diploid males (*Rabeling and Kronauer, 2013*; *Matthey-Doret et al., 2019*). Here, we use whole genome sequencing to map a sex determination locus, demonstrate that most diploid males carry a loss of heterozygosity at this locus, assess the homology of this locus with other sex determination loci, and explore whether balancing selection has shaped evolution at this locus.

## Results

### Identification of a candidate sex determination locus on chromosome 4

To map sex determination loci in *O. biroi*, we first identified males based on sexual dimorphism and distinguished diploid from haploid males based on heterozygosity at eight genetic markers (*Supplementary file 1* and *Supplementary file 2*, *Figure 2—figure supplement 1*). We then sequenced the genomes of 16 diploid males and compared them with 19 previously sequenced genomes of diploid females (*Trible et al., 2023*; *Lacy et al., 2024*) (descriptions and metadata for all genomes sequenced in this study are in *Supplementary file 3*). CSD loci must be heterozygous to trigger female development, but can be homozygous in diploid males. To identify such sites, we calculated a 'CSD index' for each single-nucleotide polymorphism (SNP), which equals zero if any female is homozygous and equals the proportion of diploid males that are homozygous if all females are heterozygous. We found a peak of the CSD index on chromosome 4, which deviated significantly from the null distribution of CSD index values (*Figure 2a*). At this peak, homozygosity levels differed significantly between males and females (*Figure 2—figure supplement 2*).

To corroborate our results, we looked for evidence of segmental losses of heterozygosity along chromosome 4. Eleven out of 16 sequenced diploid males bore segmental losses of heterozygosity on chromosome 4 (*Figure 2b*). The intersection of these LOH segments comprised a 46 kb peak at which all females were heterozygous, and most (11 out of 16) diploid males were homozygous. Under the CSD hypothesis, homozygosity for any allele at CSD loci permits male development. Consistently, *O. biroi* diploid males were homozygous for either allele at the 46 kb region on chromosome 4 (*Table 1*, *Figure 2—figure supplement 3*).

The stretches of homozygosity found in diploid males could have resulted from segmental deletions resulting from improperly repaired DNA damage or rare copy-neutral losses of heterozygosity

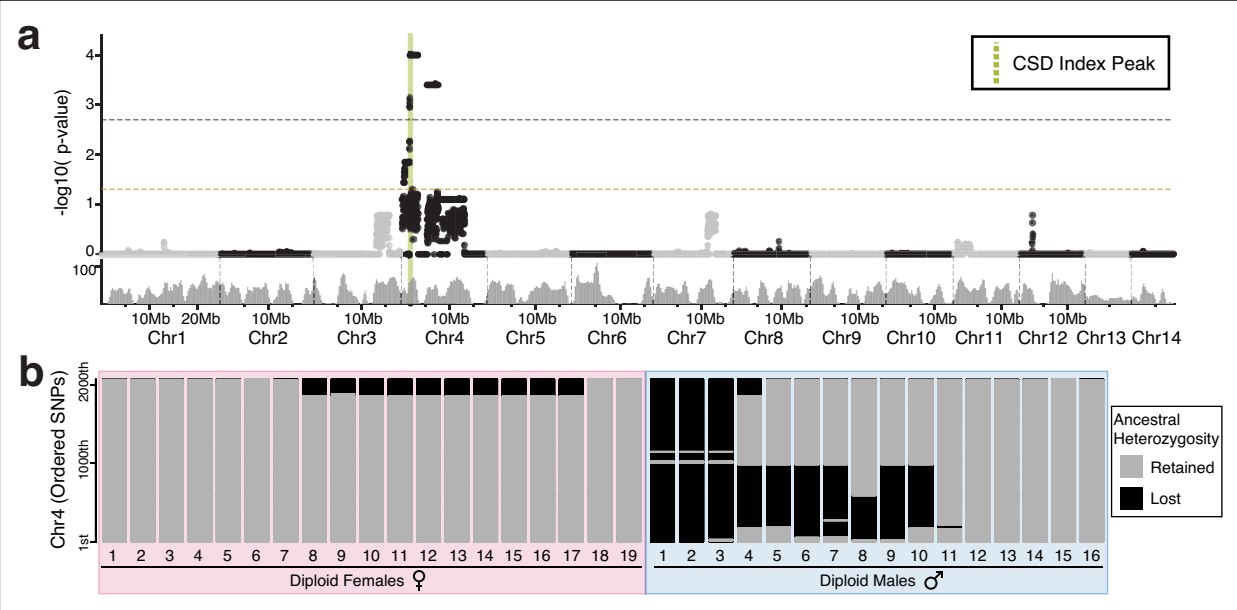

**Figure 2.** Whole genome sequencing reveals a candidate sex determination locus on chromosome 4. (**a**) Karyoplot depicting the mean complementary sex determination (CSD) index p-value in 50 kb windows with a 15 kb sliding interval, as calculated for 16 diploid males and 19 diploid females. The CSD index peak is shown as a green dotted line. The significance threshold (p=0.05) and FDR-corrected significance threshold (p=0.002) are indicated by gold and brown lines, respectively. The gray histogram shows the number of ancestrally heterozygous SNPs in 300 kb windows. (**b**) Stacked bar plots for all diploids used for mapping, with one horizontal line for each ordered putatively ancestrally heterozygous SNP on chromosome 4. For each sample, SNPs that retain ancestral heterozygosity are drawn as gray lines, whereas SNPs that have lost ancestral heterozygosity are drawn as black lines.

The online version of this article includes the following figure supplement(s) for figure 2:

**Figure supplement 1.** Genome-wide heterozygosity levels.

**Figure supplement 2.** Homozygosity levels differ between males and females at the putative complementary sex determination (CSD) locus.

**Figure supplement 3.** Diploid males result from copy-neutral losses of heterozygosity for either allele.

resulting from thelytokous parthenogenesis (*Kronauer et al., 2012*; *Oxley et al., 2014*; *Trible et al., 2023*; *Lacy et al., 2024*). To distinguish between these two scenarios, we analyzed read depth and found that runs of homozygosity were not accompanied by changes in copy number (*Figure 2— figure supplement 3*). This implies that diploid males arise when rare losses of heterozygosity, which result from crossover recombination during meiosis, span the CSD locus.

Surprisingly, 5 of the 16 diploid males were not homozygous in this 46 kb region. One possible explanation is that these individuals had mutations in other genes involved in the sex-determination pathway. We therefore inspected all unique mutations and losses of heterozygosity in these five diploid males, but did not find any in genes with annotated sex-determining functions (*Supplementary file 4*). These individuals may carry mutations or losses of heterozygosity in genes or regulatory elements with unannotated sex determination function. Alternatively, male development may have been triggered by rare stochastic perturbations to gene expression or splicing. We address these possibilities further in 'Discussion'.

**Table 1.** Number of samples heterozygous and homozygous for each allele in the 46 kb region on chromosome 4.

|  | Heterozygous | Homozygous allele 1 | Homozygous allele 2 |
| --- | --- | --- | --- |
| Diploid females | 19 | 0 | 0 |
| Diploid males | 5 | 2 | 9 |

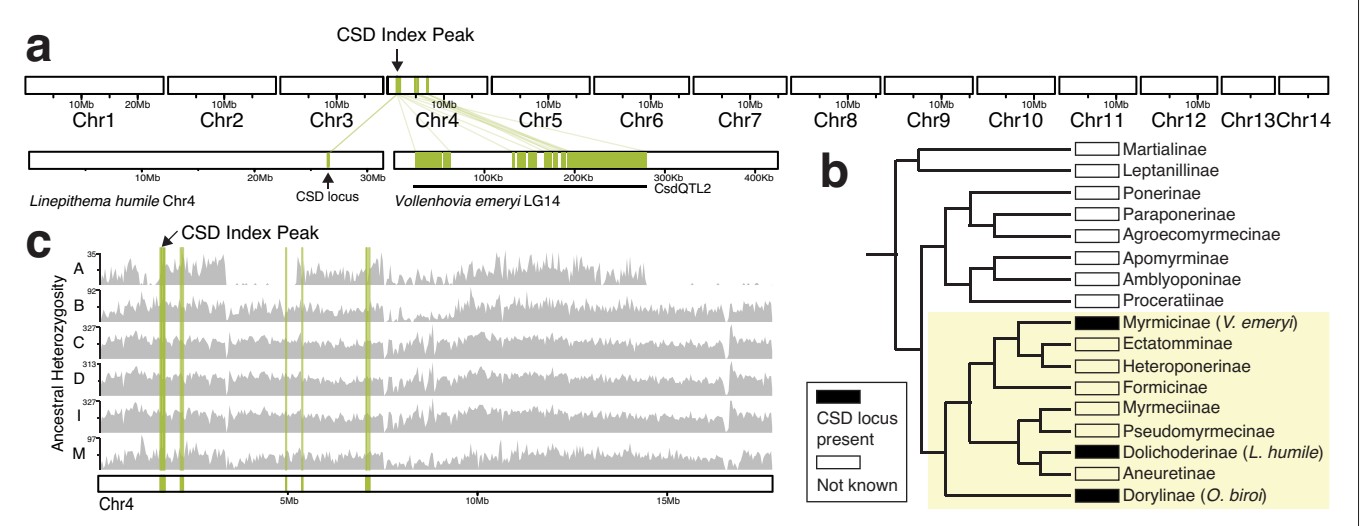

**Figure 3.** The putative sex determination locus identified in *O. biroi* is conserved across formicoid ants. (**a**) Karyoplot depicting the 14 chromosomes in the *O. biroi* genome, chromosome 4 from the *L. humile* genome (**Pan et al., 2024**), and one contig from linkage group 14 of the *V. emeryi* reference genome (**Miyakawa and Mikheyev, 2015**). Note that the plots from the different species are not drawn to scale. Homology to protein-coding genes identified within *V.emeryiCsdQTL2* is drawn in green. The location of the *O. biroi* CSD index peak is indicated. (**b**) A phylogeny of the ant subfamilies (adapted from **Borowiec et al., 2019**), with the presence of a sex determination locus homologous to the peak of our CSD index, or the absence of data shown. Note that this CSD locus has only been found in the single species indicated within parentheses for the Myrmicinae, the Dolichoderinae, and the Dorylinae. The yellow shaded background denotes the formicoid clade. (**c**) Karyoplot for *O. biroi* chromosome 4, depicting homology to *V.emeryiCsdQTL2* and ancestral heterozygosity for six different clonal lines (A, B, C, D, I, and M). Gray histograms depict the number of ancestrally heterozygous SNPs in 100 kb windows.

## *O. biroi*'s candidate CSD locus is homologous to another ant sex determination locus but not to honeybee CSD

To assess homology to CSD loci mapped in other species, we identified *O. biroi* orthologs of the genes found in other mapped sex determination loci. We found no homology between the genes within the *O. biroi* CSD index peak and any of the genes within the putative *L. fabarum* CSD loci (**Supplementary file 5**). By identifying orthologs of genes within the two sex determination QTL identified in *V. emeryi* (**Miyakawa and Mikheyev, 2015**; **Supplementary file 6**), we also identified homology to the *L. humile* CSD locus, because a subset of *V.emeryiCsdQTL2* is homologous to the CSD locus identified in *L. humile* (**Pan et al., 2024**). Homology to *V.emeryiCsdQTL2* was scattered across one arm of *O. biroi* chromosome 4, including the 46 kb CSD index peak, and homology to the *L. humile* CSD locus mapped within the *O. biroi* CSD index peak (**Figure 3a**). The *O. biroi* CSD index peak contains four protein-coding genes that are also present in *V.emeryiCsdQTL2*, including *HCF*, *COPA*, an uncharacterized gene, and *CRELD2*. Notably, the candidate CSD region in *O. biroi* also includes a large stretch of sequence with no annotated genes downstream of an uncharacterized lncRNA. LncRNAs often retain their functions and synteny with surrounding genes despite divergence in sequence homology (**Chodroff et al., 2010**; **Ulitsky et al., 2011**; **Quinn et al., 2016**), raising the possibility that, although this lncRNA has limited sequence homology with *L. humile ANTSR*, it may have a homologous function (**Pan et al., 2024**). Regardless of the underlying molecular mechanism, our data suggest that this CSD locus is conserved between *O. biroi*, *L. humile*, and *V. emeryi* (**Figure 3b**). This would mean that this locus is ancient, as these three species belong to different ant subfamilies that diverged roughly 112 million years ago (**Borowiec et al., 2025**).

We performed our genetic mapping in clonal line A because this was the only clonal line in which we found diploid males (see below). However, different clonal lines of *O. biroi* vary in which portions of the genome are ancestrally homozygous (**Oxley et al., 2014**). Therefore, it was unclear whether heterozygosity at the mapped CSD locus is required for female development in all clonal lines of *O. biroi*. To investigate this, we inspected heterozygosity in diploid females from six different clonal lines by retrieving previously published genome sequences from clonal line B (**Lacy et al., 2024**) and sequencing genomes of diploid females from four additional clonal lines (from both the native and

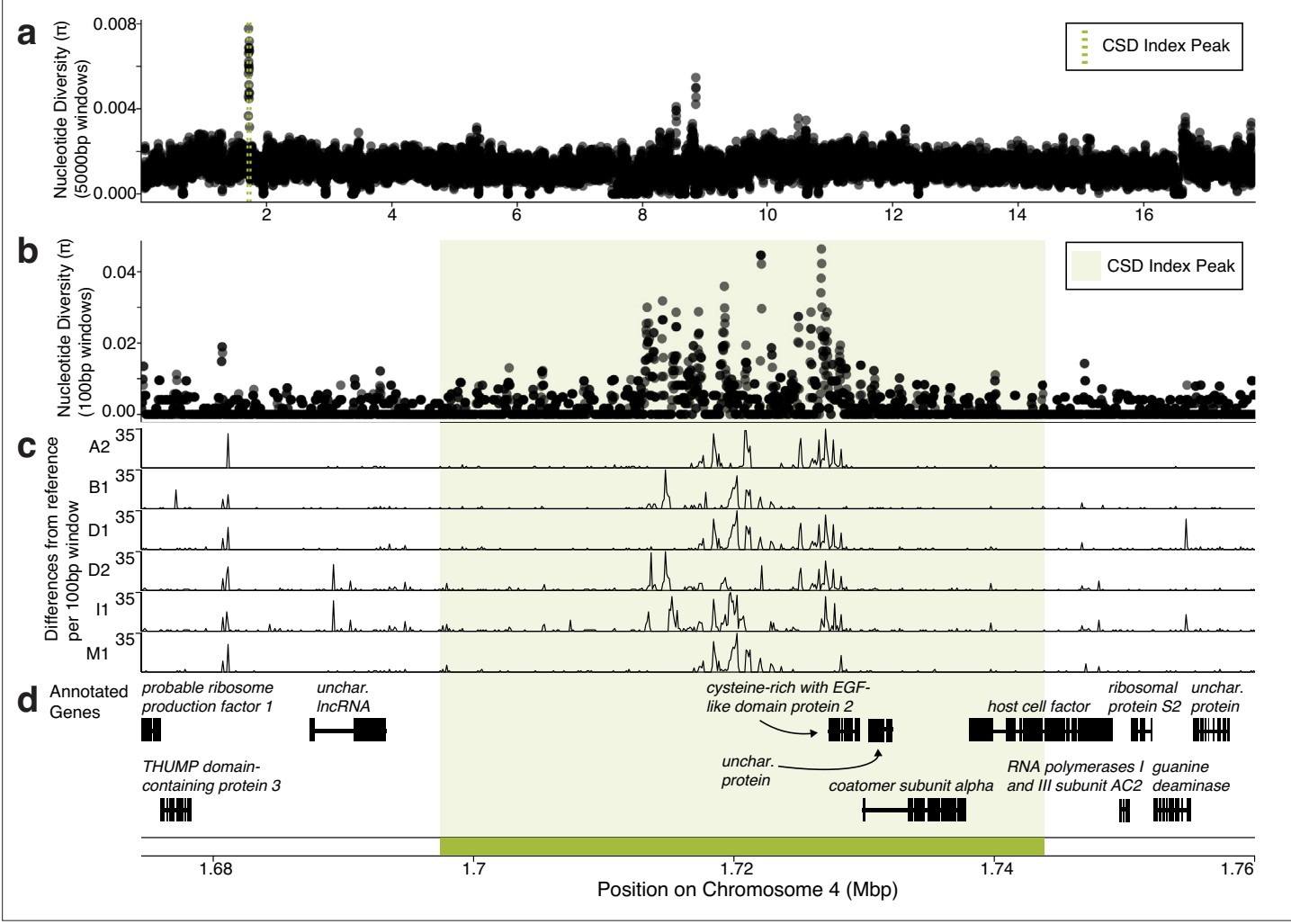

**Figure 4.** The complementary sex determination (CSD) index peak is characterized by high genetic diversity in a non-coding region. (**a, b**) Nucleotide diversity across the length of *O. biroi* chromosome 4 in 5 kb windows (step size = 1 kb) (**a**) and across the vicinity of the CSD index peak in 100 bp windows (step size = 20 bp) (**b**). (**c**) The number of differences per 100 bp window between each alternate de novo assembled allele and the reference genome allele. (**d**) Annotated genes in the vicinity of the CSD index peak. Black boxes depict exons and thin lines depict introns. The lncRNA *ANTSR* is indicated in bold. Arrows indicate the names of genes in close proximity. The CSD index peak is shown as a green dotted line in (**a**), and with green shading in (**b–d**).

The online version of this article includes the following figure supplement(s) for figure 4:

**Figure supplement 1.** Genome-wide nucleotide diversity.

**Figure supplement 2.** Sex-specific gene expression.

**Figure supplement 3.** RNAseq reads aligned to the unannotated lncRNA flanking the putative CSD locus.

invasive ranges of *O. biroi*; **Kronauer et al., 2012**; **Trible et al., 2020**). All females from all six clonal lines (including 26 diploid females from clonal line B) were heterozygous at the CSD index peak, consistent with its putative role as a CSD locus in all *O. biroi* (**Figure 3c**).

## The CSD index peak has high diversity in a non-coding region

Because CSD loci are expected to evolve under balancing selection, we expect many different alleles to exist at those loci. For example, as many as 19 *csd* alleles were found to segregate in honeybee populations (**Hasselmann et al., 2008**), and 7 alleles were found to segregate in a population of *L. humile*, despite a recent genetic bottleneck (**Pan et al., 2024**). Looking for elevated genetic diversity is impossible by only studying genomes from a single clonal line (i.e., asexual descendants of a single diploid female), so we looked across diploid females from six different clonal lines. We calculated

**Table 2.** The haplotypes present at the complementary sex determination (CSD) index peak for each studied *O. biroi* clonal line.

Because no haploid males were sequenced from clonal line C, and only one haploid male was sequenced from clonal line I, the haplotypes for these two lines remain unknown or incompletely known. We denote the unidentified haplotypes with question marks.

| Clonal line | CSD index peak haplotypes |
|---|---|
| A | A1/A2 |
| B | B1/A1 |
| C | ?/? |
| D | D1/D2 |
| I | I1/? |
| M | M1/A1 |

nucleotide diversity (π) in 5 kb sliding windows (step size = 1 kb) across the genome and found a nucleotide diversity peak that fell within the CSD index peak (*Figure 4a*, *Figure 4—figure supplement 1*), between positions 1,717,000 and 1,730,000 on chromosome 4 (*Figure 4b*).

We hypothesized that this nucleotide diversity peak resulted from many alleles found among clonal lines of *O. biroi* at this locus. Identifying these alleles requires DNA sequences of individual haplotypes, whereas short-read genome sequences of diploid females only provide diploid genotypes. To investigate allelic diversity, we retrieved previously published haploid male genome sequences from clonal lines A and B (*Lacy et al., 2024*) and sequenced whole genomes of haploid males from additional clonal lines. We assembled their genomes de novo to identify the different haplotypes found in this region. In addition to the haplotype found in the reference genome, we identified six haplotypes (or 'alleles') in the five clonal lines for which we sequenced at least one haploid male genome (*Figure 4c*, *Table 2*). These alleles differ substantially in their DNA sequences in the region with high nucleotide diversity (*Figure 4c*).

To investigate which genetic elements might act to determine sex in a heterozygosity-dependent manner, we inspected all annotated genes in the CSD index peak (*Figure 4d*). Within this region, none of the protein-coding genes contained substantial heterozygosity that affected amino acid identity (*Supplementary file 7*). To determine whether any genes or exons were missing from our annotation, we performed RNA sequencing of embryos and several other life stages (including female and male adults) using several different technologies (see 'Materials and methods'). Briefly, we used long-read RNA sequencing (PacBio IsoSeq) to extend gene models and used several different library preparation methods followed by standard short-read (Illumina) sequencing: standard poly-A tail library prep to capture mRNAs, ribosomal RNA depletion to capture lncRNAs that are not poly-adenylated, and small RNA sequencing to annotate miRNAs, endo-siRNAs, and putative piRNAs. Although this improved the *O. biroi* genome annotation by adding new genes and improving existing gene models (*Supplementary file 8* and *Supplementary file 9*), no new genes or exons were identified near the CSD index peak.

To look for sex-biased gene expression and splicing, we performed differential gene expression and differential exon usage analysis between the transcriptomes of three male and three female early-stage (white) pupae. We used pupae for this experiment because this is the earliest life stage at which males and females can be readily distinguished. Primary sex determination signals act early during embryonic development (*Sawanth et al., 2016*) and thus need not be differentially expressed later in development. However, these genes can still be expressed as late as the pupal stage (*Beye et al., 2003*). Many genes were differentially expressed between males and females (24.6% of expressed genes) or had one or more differentially used exons, including four genes near the CSD index peak (LOC105285605, LOC105283850, LOC105283849, and LOC105283844) (*Supplementary file 10* and *Supplementary file 11*, *Figure 4—figure supplement 2*). However, none of these overlapped with the region of elevated genetic diversity. Very few reads from the male and female pupae samples aligned to the lncRNA, and although more reads aligned in RNA-seq libraries from eggs prepared using ribosomal RNA depletion, these data are insufficient for a rigorous assessment of sex-specific

**Table 3.** Number of haploid and diploid males sampled from different *O. biroi* clonal lines.

| Clonal line | Haploid males | Diploid males |
|---|---|---|
| A | 92 | 22 |
| B | 78 | 0 |
| C | 6 | 0 |
| D | 4 | 0 |
| I | 1 | 0 |
| M | 2 | 0 |

expression at this locus (*Figure 4—figure supplement 3*). Therefore, it remains unclear whether this lncRNA plays a role in sex determination in *O. biroi*.

The peak of nucleotide diversity minimally affects amino acid sequences and is primarily located in a non-coding genomic region (*Figure 4b–d*). Thus, a non-coding genetic element within this region may instruct sex determination in *O. biroi* in a heterozygosity-dependent manner. This is similar to *L. humile*, where the mapped CSD locus had high variability in non-coding and non-exonic regions (*Pan et al., 2024*). RNA interference of the nearby lncRNA, *ANTSR*, led to male-specific splicing of *tra*, raising the possibility that heterozygosity or homozygosity in the non-exonic and non-coding *L. humile* CSD locus affects the expression level of *ANTSR*, which instructs sex-specific splicing of *tra* (*Pan et al., 2024*). Because the peak of nucleotide diversity within our CSD index peak minimally affects protein-coding sequences and the region is closely linked to a lncRNA that is putatively an ortholog of *L. humile ANTSR*, our data suggest that the mechanism of CSD in *O. biroi* may be conserved with *L. humile*.

## Diploid male production differs across *O. biroi* clonal lines

Over the course of our study, we sampled many males from clonal lines A and B, and a few males from four additional clonal lines. Despite the fact that we maintain more colonies of clonal line B than of clonal line A in the lab, all the diploid males we detected came from clonal line A. We, therefore, looked back at data from previously genotyped males and found that this pattern held across a previous study (*Kronauer et al., 2012*). In total, from clonal line A, 22 out of 114 males were diploid, whereas all 78 males genotyped from clonal line B were haploid (*Table 3*). This disparity is statistically significant ($\chi^2$=15.4, df = 1, p<0.0001), suggesting that the two clonal lines differ in their propensity to produce diploid males.

One possible explanation for this phenomenon could be that *O. biroi* has multi-locus CSD (*Figure 1b*), but that clonal line A is ancestrally homozygous for all but one of the loci, whereas clonal line B retains heterozygosity at two or more loci. Because multi-locus CSD requires homozygosity at all sex determination loci for diploids to develop as males, in this scenario, two or more losses of heterozygosity would be needed for diploid males to develop in clonal line B. In contrast, only one loss of heterozygosity would be required in clonal line A, which we used for our genetic mapping. However, we identified no additional candidate loci in our mapping study, and other possibilities are equally likely, including that clonal line A is more prone to loss of heterozygosity, or is more permissive to diploid male development than other clonal lines.

## *O. biroi* lacks a *tra*-containing CSD locus

A second QTL region identified in *V. emeryi* (*V.emeryiCsdQTL1*) contains two closely linked *tra* homologs, similar to the closely linked honeybee *tra* homologs, *csd* and *fem* (*Miyakawa and Mikheyev, 2015*). This, along with the discovery of duplicated *tra* homologs that undergo concerted evolution in bumblebees and ants (*Schmieder et al., 2012*; *Privman et al., 2013*), has led to the hypothesis that the function of *tra* homologs as CSD loci is conserved with the *csd*-containing region of honeybees (*Schmieder et al., 2012*; *Miyakawa and Mikheyev, 2015*). However, other work has suggested that *tra* duplications occurred independently in honeybees, bumblebees, and ants (*Hasselmann et al., 2008*; *Koch et al., 2014*), and it remains to be demonstrated that either of these *tra* homologs acts as a primary CSD signal in *V. emeryi*. To test whether the *tra* homolog-containing QTL (*V.emeryiCsdQTL1*)

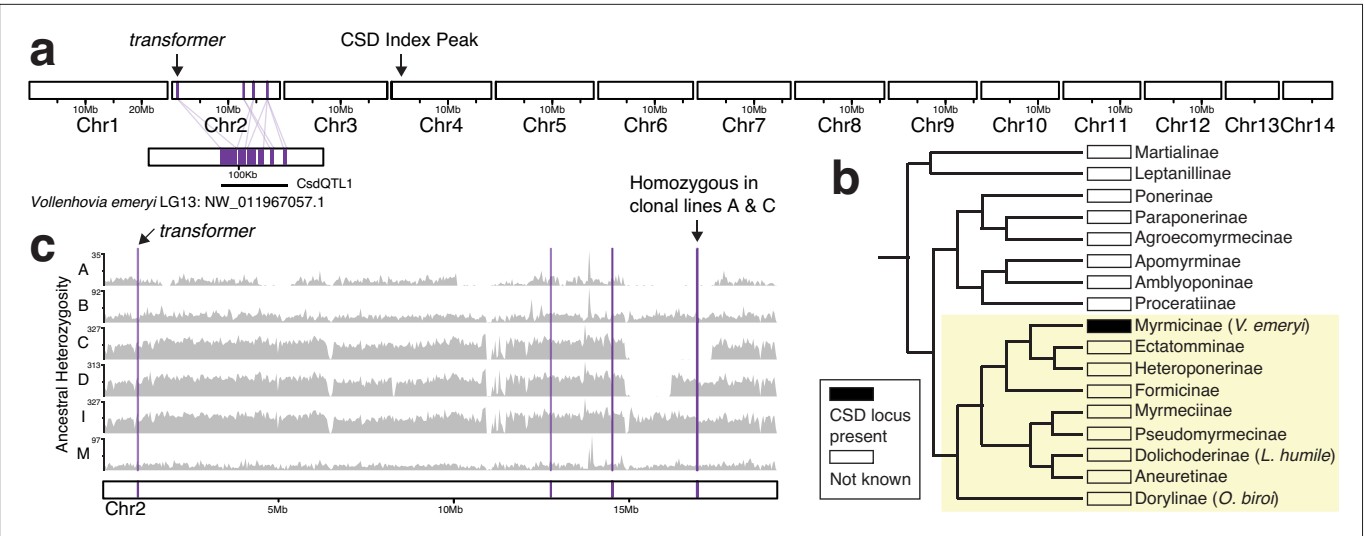

**Figure 5.** Whether a second, *tra*-containing complementary sex determination (CSD) QTL from *V. emeryi* is conserved across ants remains ambiguous. (**a**) Karyoplot depicting the 14 chromosomes in the *O. biroi* genome and one contig from linkage group 13 of the *V. emeryi* reference genome. Homology to protein-coding genes identified within *V.emeryiCsdQTL1* is drawn in purple. The locations of the *O. biroi* CSD index peak and the homolog of *transformer* are indicated. (**b**) A phylogeny of the ant subfamilies (adapted from *Borowiec et al., 2019*), with the presence or absence of this second putative sex determination locus, or the absence of data shown. Note that the *tra*-containing candidate CSD locus has only been found in a single species (*V. emeryi*), rather than in all Myrmicinae. The yellow shaded background denotes the formicoid clade. (**c**) Karyoplot for *O. biroi* chromosome 2, depicting homology to *V.emeryiCsdQTL1* and ancestral heterozygosity for each clonal line. Gray histograms depict the number of ancestrally heterozygous SNPs in 100 kb windows. The region of homology that is ancestrally homozygous in clonal lines A and C is labeled.

The online version of this article includes the following figure supplement(s) for figure 5:

**Figure supplement 1.** The *O. biroi* transformer homolog is sex-specifically spliced.

might be involved in CSD in *O. biroi*, we searched for homology to this QTL in the *O. biroi* genome and identified several regions scattered across *O. biroi* chromosome 2 (*Figure 5a*). However, we did not find a CSD index peak on chromosome 2, and no sex determination locus with homology to the *V. emeryi* QTL was found in *L. humile* (*Pan et al., 2024*). Thus, the function of *V.emeryiCsdQTL1* as a primary sex determination signal might have evolved after this lineage separated from other ants (*Figure 5b*). Alternatively, the sex determination activity of *V.emeryiCsdQTL1* could be ancestral to these three ant species and independently lost in *O. biroi* (at least in clonal line A) and *L. humile*.

Intriguingly, while clonal lines B, D, I, and M retain heterozygosity at all sites with homology to *V.emeryiQTL1* (*Figure 5c*), clonal lines A and C are homozygous in a region homologous to a portion of *V.emeryiQTL1*. We note, however, that heterozygosity in this region is limited, without elevated nucleotide diversity (*Figure 4—figure supplement 1*), and there are only two nonsynonymous heterozygous variants in this region, with clonal line B conspicuously lacking any nonsynonymous heterozygosity. Therefore, it seems unlikely that this region bears a CSD locus in *O. biroi*. We also note that all clonal lines are heterozygous in the region containing *O. biroi*'s *tra* ortholog (*Figure 5c*), and that this gene is sex-specifically spliced (*Figure 5—figure supplement 1*), similar to *tra* homologs in other Holometabola (*Hasselmann et al., 2008*; *Miyakawa and Miyakawa, 2023*). This contrasts with the *tra-like* gene in *V. emeryi*, which, like *A. mellifera csd*, is not sex-specifically spliced (*Miyakawa and Miyakawa, 2023*).

## Discussion

Genetically mapping sex determination loci in diverse taxa is key to understanding how sex determination evolves. Here, we investigated the mode of sex determination in the clonal raider ant, *O. biroi*. First, we mapped a candidate sex determination locus that was heterozygous in all females, but homozygous for either allele in most diploid males (*Figure 2*). Next, we showed that most diploid males bear rare losses of heterozygosity that arise during thelytokous parthenogenesis (*Figure 2*). We then showed that this locus is homologous to a sex determination locus found in two other ant

species—*V. emeryi* and *L. humile* (**Figure 3**), suggesting that this mechanism of sex determination may be conserved among formicoid ants. By sequencing whole genomes of haploid males and diploid females from other clonal lines of *O. biroi,* we showed that the mapped locus corresponds to a region with high genetic diversity, following the expectation that CSD loci should evolve under balancing selection (**Figure 4**). This diversity peaks in a noncoding genomic region, and we found limited evidence for functional heterozygosity in nearby protein-coding sequences, suggesting that hetero-zygosity in this noncoding sequence may trigger female development. Finally, we demonstrated that *O. biroi* CSD in clonal line A does not map to the *tra*-containing sex determination locus previously identified in *V. emeryi*, but it remains unclear whether *O. biroi* ancestrally has single-locus or multi-locus CSD (**Figure 5**).

The presence of diploid males in many ant taxa previously led to the conclusion that most ants employ CSD (**van Wilgenburg et al., 2006**). Along with *L. humile* (**Pan et al., 2024**) and *V. emeryi* (**Miyakawa and Mikheyev, 2015**), *O. biroi* is now one of three ant species for which heterozygosity-dependent female development has been demonstrated at mapped CSD loci. Our mapped locus appears to be homologous to *V.emeryiCsdQTL2* and to the locus identified in *L. humile* (**Figure 3**). As was found in *L. humile*, this locus colocalizes with a peak of nucleotide diversity (**Figure 4**), with seven alleles found among five clonal lines (**Figure 4c**, **Table 2**), consistent with the expectation that CSD loci should harbor many alleles due to evolution under balancing selection. This implies that the role of this locus in CSD evolved before the divergence of the Dorylinae from the other formicoid ants roughly 112 million years ago (**Borowiec et al., 2025**). Although synteny analyses raised the possibility that this locus arose early within the evolution of the Aculeata (**Pan et al., 2024**), genetic mapping studies of additional representative species of different ant subfamilies (especially in the poneroid clade and the Leptanillinae) and across other Aculeata will be required to determine when the func-tion of this locus in CSD originated and how it is distributed across taxa.

In *L. humile*, a lncRNA (*ANTSR*) closely linked to this locus plays a role in sex determination (**Pan et al., 2024**). Although we observed some expression of the putatively homologous lncRNA in *O. biroi* eggs of unknown sex, we were unable to assess sex-specific expression. The reason is that males in *O. biroi* are exceedingly rare and morphologically only identifiable at the pupal stage, well after primary sex determination signals are transduced to self-sustaining sex-specific splicing of down-stream transcription factors during the early stages of embryonic development (**Gempe and Beye, 2011**). Therefore, we likely missed differential gene expression relevant to sex determination that occurs before the pupal stage.

We did not find a peak of the CSD index at the *O. biroi tra* homolog (**Figure 5**), indicating that, as in *L. humile*, the mode of CSD is not homologous to the system in *A. mellifera* mediated by the *transformer* homologs *csd* and *fem*. This was previously suspected for *O. biroi* due to the presence of only a single *tra* homolog (**Oxley et al., 2014**) rather than the two closely linked *tra* homologs found in other ants (**Privman et al., 2013**). In *V. emeryi*, however, one of the two Csd QTLs (*V.emery-iCsdQTL1*) spans the *tra* locus, raising the possibility that the molecular mechanism of CSD might, in part, be conserved between *A. mellifera* and *V. emeryi* (**Miyakawa and Mikheyev, 2015**). It has not been demonstrated that either of *V. emeryi*'s *tra* homologs acts as a primary sex determination signal. Our results raise the possibility that the role of *V.emeryiCsdQTL1* as a primary sex determi-nation signal evolved independently of *A. mellifera csd* and after *V. emeryi* diverged from *L. humile*. However, further investigations into the mechanisms of sex determination in *V. emeryi* and other ants are required to reconstruct the evolutionary history of *V.emeryiCsdQTL1*'s function in hymenopteran sex determination.

Curiously, diploid males were only produced in clonal line A, even though many males were collected from other clonal lines (**Table 3**). One possible explanation is that *O. biroi* has multi-locus CSD, and clonal line A is ancestrally homozygous for all but one of the loci. Multi-locus CSD has been suggested to limit the extent of diploid male production in asexual species under some circum-stances (**Vorburger, 2014**; **Matthey-Doret et al., 2019**). Clonal line A is ancestrally homozygous for a portion of the region homologous to the second *V. emeryi* QTL, although it retains heterozygosity in the region overlapping *tra* (**Figure 5c**). However, this region lacks elevated genetic diversity and functional heterozygosity, making it a poor candidate for a second, undetected, CSD locus. Further study would be required to determine whether *O. biroi* has a second CSD locus. However, there may be other explanations for diploid males being found exclusively in clonal line A. For example, clonal

line A could be more likely to lose heterozygosity than other clonal lines, or diploid males in clonal line A could be more likely to develop and survive. Currently, we have no data to distinguish between these hypotheses.

Another peculiarity of our study is that 5 of the 16 sequenced diploid males retained heterozygosity at the mapped CSD locus. One possible explanation for this would be that our mapped locus is, in fact, a false positive. However, this seems unlikely given that (1) those losses of heterozygosity occurred independently in multiple genetic backgrounds within clonal line A (*Figure 2—figure supplement 3*), (2) a peak of nucleotide diversity co-occurs with the mapped locus (*Figure 4*), and (3) this locus is homologous to CSD loci mapped in two other ant species (*Figure 3*). Stochastic gene expression or splicing fluctuations may have caused these individuals to develop as males despite having "female genotypes". Certain allelic combinations of *A. mellifera csd* trigger female development with incomplete penetrance (*Beye et al., 2013*), and the allelic combination found in clonal line A might similarly be permissive to occasional male development. Alternatively, these diploid males could result from mutations in or losses of heterozygosity at different genetic loci. We note that it might be easier to find rare stochastically produced diploid males in laboratory colonies of *O. biroi* than in sexual species. This is because genetically encoded diploid males only result from loss of heterozygosity, which is rare in *O. biroi* (*Oxley et al., 2014*; *Lacy et al., 2024*). By contrast, in sexually reproducing populations, diploid males homozygous for sex determination loci will arise much more frequently due to matings between individuals that share an allele at the sex determination locus (this effect is exacerbated in invasive species like *L. humile* that have undergone recent genetic bottlenecks). Thus, if stochastically produced diploid males occur at very low baseline levels in ants, such males will rarely be detected in sexual species. In *O. biroi*, however, they could make up a substantial proportion of diploid males, simply because diploid males caused by homozygosity at the CSD locus are likewise extremely rare.

Finally, we note that heterozygosity-dependent female development, a fundamental tenet of CSD, has major implications for the evolution of breeding systems and reproductive modes. Due to the high fitness cost of producing sterile diploid males, species with CSD should obligatorily outbreed or have other mechanisms of maintaining heterozygosity (*van Wilgenburg et al., 2006*). In this study, most diploid males resulted from rare crossover-associated losses of heterozygosity that spanned the putative CSD locus (the shortest loss of heterozygosity at the mapped locus was 46 kb and, therefore, likely resulted from crossover recombination rather than gene conversion). Intriguingly, in many taxa, asexual lineages reproduce via mechanisms that lead to complete homozygosity in a single generation (*Suomalainen et al., 1987*; *Ma and Schwander, 2017*). However, among the social Hymenoptera (and other Vespoids), the known modes of asexual reproduction all maintain at least some heterozygosity (*Rabeling and Kronauer, 2013*; *Ma and Schwander, 2017*). For formicoid ants, bees, and other taxa with CSD, asexual reproduction with high rates of heterozygosity loss would incur a steep fitness penalty due to the production of sterile diploid males. This may help explain the unusual inheritance system recently described in *O. biroi*, where following crossover recombination, reciprocally recombined chromatids are faithfully co-inherited so that heterozygosity is rarely lost (*Lacy et al., 2024*).

How this ancient putative CSD locus triggers female development in a heterozygosity-dependent manner remains unknown. In *L. humile*, RNAi experiments suggested that the expression level of the lncRNA *ANTSR* affects sex-specific splicing of *tra*, even though heterozygosity in the exons of *ANTSR* is not required for this effect (*Pan et al., 2024*). Thus, the heterozygosity-dependent function of the candidate CSD locus may depend on local regulatory interactions between homologous chromosomes, similar to a phenomenon known as transvection (*Duncan, 2002*). To better understand how heterozygosity induces female development in ants, future studies will need to fine-map and functionally characterize this causal locus.

## Materials and methods
### Software versions
The versions of all software used for analyses are provided in *Supplementary file 12*.

### Identifying male ploidy
Clonal raider ant colonies are composed of female ants that reproduce asexually via a type of thelytokous parthenogenesis termed central fusion automixis (*Oxley et al., 2014*). Males are produced

only sporadically (*Oxley et al., 2014*). Females of this species are diploid, whereas males can either be haploid (presumably arising via a failure of central fusion following female meiosis) or diploid (presumably arising due to rare losses of heterozygosity at CSD loci). Males are easily recognizable due to substantial sexual dimorphism (males have eyes and wings, whereas females are blind and flightless, and males also have different body shape and more darkly pigmented cuticles than females). We sampled them opportunistically whenever they were observed in colonies. We distinguished diploid males from haploid males based on heterozygosity at several unlinked genetic markers (*Supplementary file 1*). For this, we disrupted one leg from each male using a QIAGEN TissueLyser II and extracted DNA using QIAGEN's QIAmp DNA Micro Kit. We then PCR-amplified each marker and genotyped using Sanger sequencing (for most markers) or via gel electrophoresis either after restriction digestion (for two markers) or directly after amplification (for one marker at which the different alleles are different sizes). Details of the genotyping methods can be found in *Supplementary file 1*.

## Genome sequencing and preliminary analysis

For each library, we disrupted an individual ant with a metal bead and QIAGEN's TissueLyser II and extracted DNA with QIAGEN's QIAmp DNA Micro Kit. We prepared Nextera Flex Illumina DNA sequencing libraries for 16 diploid males from clonal line A, six diploid females from different clonal lines, and seven haploid males from different clonal lines. We sequenced these libraries using an Illumina NovaSeq 6000. We also accessed libraries for individual genomes from previous studies. The information for all DNA sequencing libraries used in this study can be found in *Supplementary file 3*.

We trimmed reads using Trimmomatic 0.36 (*Bolger et al., 2014*), aligned them to the *O. biroi* reference genome (*McKenzie and Kronauer, 2018*) (Obir_v5.4, GenBank assembly accession: GCA_003672135.1) using bwa mem (*Li, 2013*), and then sorted, deduplicated, and indexed using picard (http://broadinstitute.github.io/picard/). We called variants using GATK HaplotypeCaller (version 4.2) (*Poplin et al., 2018*). To exclude falsely collapsed regions in the reference genome, we filtered against sites found to be heterozygous in haploid males. We also screened out any variants for which putatively heterozygous samples had a proportionate minor allelic depth less than 0.25 and/or for which putatively homozygous samples had a minor allelic depth greater than zero. We performed custom filtering of variants and calculated statistics using pyvcf 0.6.8 (*Casbon, 2015*), and performed sliding window analyses using pybedtools 0.9.0 (*Dale et al., 2011*).

## CSD index mapping

CSD loci are heterozygous in females but can be homozygous in diploid males. Thus, traditional association mapping cannot identify CSD loci because particular alleles do not trigger female development. Therefore, to map candidate CSD loci, we used a 'CSD Index', which equals zero if any female is homozygous, but if all females are heterozygous, it equals the proportion of diploid males that are homozygous. Before calculating this index, we had to identify ancestrally heterozygous sites. All individuals within an *O. biroi* clonal line are descended asexually from a single diploid female ancestor, and CSD loci would have been heterozygous in that common ancestor. Putatively ancestrally heterozygous variants are those for which at least one diploid individual is heterozygous, and all other diploid individuals are either heterozygous for the same two alleles, or homozygous for one of the two alleles for which other diploid(s) are heterozygous (*Oxley et al., 2014*; *Lacy et al., 2024*). After identifying putatively ancestrally heterozygous SNPs, we calculated the CSD index for each SNP.

To assess the likelihood that our observed CSD index values were due to chance (false positives), we randomly shuffled the identities of males and females to generate a null distribution of CSD index values for each SNP. In doing so, we maintained the underlying genetic structure by only shuffling sex labels within colonies. We computed p-values by calculating the proportion of permutations with values greater than or equal to the observed CSD index. To correct for multiple testing and control the false discovery rate across the genome-wide analysis, we applied the Benjamini–Hochberg false discovery rate correction and then recorded the mean CSD index value in sliding windows.

To directly evaluate the statistical association between homozygosity and sex, we applied Fisher's exact test to assess whether SNP homozygosity frequencies differed significantly between males and females and used the Benjamini–Hochberg false discovery rate correction to adjust p-values.

## Losses of heterozygosity

We identified losses of heterozygosity as ancestrally heterozygous SNPs that had become homozygous in a given sample. To assess whether homozygosity for either allele at the mapped region permitted male development, we randomly chose a reference haploid male and assigned allelic identity based on whether a focal sample was homozygous for the same allele or the alternate allele to that possessed by the reference haploid male. To clearly illustrate losses of heterozygosity, we used the ordinal position of SNPs rather than the position of SNPs in the reference genome assembly.

Homozygosity in whole genome sequencing data can result from deletions, for which the affected region would have half the read depth of the genome-wide average, or copy-neutral losses of heterozygosity (which result rarely from meiotic recombination and central fusion parthenogenesis; *Kronauer et al., 2012*; *Rabeling and Kronauer, 2013*; *Oxley et al., 2014*; *Lacy et al., 2024*). To determine which of these processes caused homozygosity in diploid males, we identified runs of homozygosity using "bcftools roh" (*Narasimhan et al., 2016*), obtained read depths at all SNPs that passed filtering within these regions using "samtools depth –aa" (*Danecek et al., 2021*) and then normalized by dividing by the genome-wide median read depth. For samples without losses of heterozygosity on chromosome 4, we obtained normalized read depth at all SNPs that passed filtering on chromosome 4.

## Homology between CSD loci across species

To determine whether our mapped CSD locus bore homology to those identified in other ant species, we used OrthoFinder (*Emms and Kelly, 2019*) to identify orthologs of the genes present in the two CSD QTL in *V. emeryi* (*Miyakawa and Mikheyev, 2015*). The locus identified in *L. humile* (*Pan et al., 2024*) is homologous to a subset of one of the *V. emeryi* QTL. In this process, we noticed that one of the genes found within our mapped CSD locus (*cysteine-rich with EGF-like domain protein 2, or CRELD2*) was also present in a second copy in a contig that was erroneously duplicated during the genome assembly process (Chr4:21571–39095). We masked this contig and realigned sequencing data for all subsequent analyses.

## Balancing selection

Because CSD loci are expected to evolve under negative frequency-dependent (balancing) selection, many different alleles should segregate in populations. To assess this, we used scikit-allel (*Miles et al., 2019*) to calculate nucleotide diversity across six different clonal lines and reported nucleotide diversity in sliding windows.

From inspecting DNA sequencing alignments to the reference genome, it became clear that the high genetic variation within the mapped CSD locus likely resulted from several highly differentiated haplotypes in this region. To investigate this further, we sequenced whole genomes of haploid males from clonal lines other than A and B. We assembled the genomes of each haploid male de novo using SPAdes (*Bankevich et al., 2012*; *Prjibelski et al., 2020*), found the contig bearing homology to the mapped CSD locus in the reference genome assembly using blastn (*Camacho et al., 2009*), manually scaffolded contigs as needed, and pairwise aligned each alternate allele to the reference genome using nucmer and mummer (*Kurtz et al., 2004*) to identify differences between alleles.

## RNAseq for annotation improvements

To avoid missing any previously unannotated genes in the mapped CSD locus region, we performed RNA sequencing following various approaches. To start, we extracted total RNA from multiple samples, including separate pooled tissue from several different life stages (eggs, young larvae, fourth instar larvae, prepupae, pupae, and adults), different adult tissues (antennae, heads, thoraces, legs, abdomens), and individual male and female pupae using a TriZol RNA extraction method followed by precipitation in isopropanol at –20°C overnight. We used the different life stage extractions for long read RNA-seq, rRNA depletion RNA-seq, and small RNA-seq. We used the different adult tissues for long read RNA-seq, and the individual male and female pupae for mRNA-seq. Long read RNA-seq was performed on a PacBio Sequel II, and all other sequencing was performed on an Illumina NovaSeq 6000. For rRNA depletion, we used Illumina's TruSeq Stranded Total RNA RiboZero kit. For small RNA-seq, we prepared libraries using Perkin Elmer's NEXTFLEX Small RNA-Seq Kit v3 for Illumina Platforms.

## Annotation improvements: Long read transcriptome assembly

The GenBank reference genome (GCA_003672135) for *O. biroi* (Obir_v5.4) was retrieved from NCBI in FASTA format alongside the GenBank (GCA_003672135) and RefSeq (GCF_003672135.1) transcriptomes in GTF format. Additionally, a curated RefSeq annotation was retrieved from Zenodo (https://doi.org/10.5281/zenodo.10079884) in GTF format. RefSeq assemblies were chromosome name-mapped to GenBank chromosome names using the Bioconductor GenomeInfoDb package (*Arora et al., 2024*) and the chromosome name maps available in NCBI.

The PacBio toolset's pbmm2 (*Pacific Biosciences, 2023*), a PacBio wrapper for Minimap2 (*Li, 2018*), was used for indexing and alignment of pooled full-length, non-concatemer long reads to the GenBank reference genome using the parameters "--preset ISOSEQ -j 8". Following the generation of aligned long reads in BAM format, Stringtie2 (*Kovaka et al., 2019*) was used with parameters "-L" for long read transcriptome assembly, and the curated RefSeq annotation was supplied as a guide for assembly using the "-G" parameter. The resulting gene models were exported for further analysis and integration as a gzipped GTF file.

## Annotation improvements: Short read transcriptome assembly

For Illumina RNA-seq data, FastQ quality control was performed using Rfastp (*Wang and Carroll, 2024*). Indexing and alignment of short reads to the GenBank genome was performed using Hisat2 (*Kim et al., 2019*) with default parameters, and samtools was used to generate sorted and indexed BAM files for further analysis. Stringtie2 was then used to assemble transcriptomes using the default parameters. Gene models were then combined across replicates using Stringtie2's merge function with parameters "-F 0.5 -T 2 -f 0.1" and were exported for further analysis and integration as a gzipped GTF file.

## Annotation improvements: Integrating short and long read transcriptomes with the RefSeq assembly

Integration of long read and short read generated gene models with the established and curated RefSeq transcriptome was performed using custom scripts in R available from GitHub (https://github.com/RockefellerUniversity/Obiroi_GeneModels_2025, copy archived at *Carroll and Wang, 2025*) and described here.

First, conflicting gene models where single long read genes map to multiple RefSeq genes and vice versa were identified. As generated long read gene models are supported by multiple full-length, non-concatemer long read alignments, attention was focused on the 720 instances of multiple long read generated genes overlapping single RefSeq genes.

Single exon long read genes were removed as these are often found to be artifacts from PacBio sequencing. Following this, long read gene models that split RefSeq gene models but matched with GenBank gene models were accepted with the corresponding RefSeq models rejected.

Next, RefSeq gene models that conflicted within themselves were removed. This includes genes that were overlapping in their exons but marked as separate genes.

After tidying RefSeq gene models, long read gene models that split RefSeq gene models but matched with short read gene models were accepted, with the corresponding RefSeq models rejected. After this gene model polishing, only 132 gene models were left where multiple long read generated genes overlapped single RefSeq genes.

To potentially rescue internally primed transcripts caused by long stretches of genomic polyAs, long read transcripts were assessed for polyA nucleotide content downstream from their endpoints, and these distributions were visualized to identify cut-offs for putative internally primed transcripts. Transcripts with at least 70% as 10 bp downstream of the last exon or with a polyA stretch >4 in 10 bp of the last exons were labeled as putative internally primed transcripts. For putative internally primed transcripts, which also split RefSeq genes, short read gene models were used to guide the rescued models when both short read and RefSeq agreed on a single gene model.

Finally, for the remaining long read gene models that split RefSeq gene models, gaps between long read gene models were assessed in the short read and RefSeq gene models. If agreement in exon and junction structure was found between short read and RefSeq gene models, the long read gene models were rejected. This left a final 27 genes where long read gene models split RefSeq gene

models, and these were manually assessed in IGV alongside short read and long read alignments to reject or accept.

With a final set of long read gene models, all RefSeq and short read transcripts that fell within single genes were then added to the long read transcriptome to create an integrated transcriptome. Duplicate transcripts were then removed from the integrated transcriptome, and transcripts and genes were annotated for their sources (long read, short read, RefSeq) and any overlap with the established RefSeq, GenBank, and Ensembl gene models included. Following the creation of the integrated gene models, SQANTI3 (*Pardo-Palacios et al., 2024*) was used to provide a final corrected set of gene models as well as additional QC, as described below.

## Assessment of annotation quality and identifying novel genes, transcripts, and exons

SQANTI3 was run with default parameters on the integrated transcriptome against the curated RefSeq and GenBank gene models to generate QC and descriptive statistics on novel and non-canonical splicing events. Gffcompare software (*Pertea and Pertea, 2020*) was run against the curated RefSeq and GenBank gene models to provide statistics on novel exons, transcripts, and genes.

BUSCO (*Simão et al., 2015*) was run on the integrated, curated RefSeq and GenBank gene models with the parameters "-l insecta_odb10 -m tran" to provide measures of transcriptome completeness based on a set of core gene models. All these metrics were summarized into tables using custom R scripts available on GitHub (https://github.com/RockefellerUniversity/Obiroi_GeneModels_2025, copy archived at *Carroll and Wang, 2025*).

## Annotation of CDS and functional annotation of genes

Annotation of coding regions was performed using TransDecoder (*Haas, 2023*) to add CDS information to our integrated gene models GTF. TransDecoder was run with default parameters except for providing the "--single_best_only" to the TransDecoder.Predict function to get only one prediction per transcript. CDS information was merged with SQANTI3's GeneMark CDS annotation for a final set of CDSs. With this coding information, gene models were read into R, translated into protein sequences using the BioStrings package, and protein per transcript sequences exported as FASTA format for use in downstream tools. For functional annotation, both eggNOG (*Huerta-Cepas et al., 2019*) and InterProScan (*Jones et al., 2014*) were used using default parameters with our protein sequences FASTA to provide predictions for functional annotations of genes including their putative GO, KEGG, PFAM, and Panther categories. Annotations were summarized to gene levels from transcript level annotation using R.

## Annotation of lncRNAs

FEELnc (*Wucher et al., 2017*) was used to identify putative lncRNAs within our final integrated gene models. First, the FEELnc_filter.pl script was run with default parameters, and then the FEELnc_codpot.pl script was run with parameters "--mode=shuffle" to generate our putative set of lncRNAs for annotation of our integrated gene model GTF.

## Annotation of miRNAs

Small RNA-seq reads were adaptor clipped for a "TGGAATTCTCGGGTGCCAAGG" adapter using CLIPflexR (*Rozen-Gagnon et al., 2021*) and aligned to the genome using miRDeep2's mapper.pl script with parameters "-c -j -m -l 18" (*Friedländer et al., 2012*). Aligned data were then processed using miRDeep2's miRDeep2.pl script with a FASTA of known *Drosophila* miRNAs retrieved from miRBase (*Kozomara et al., 2019*). The resulting annotated miRNA genomic locations were merged across samples and added into our integrated gene model GTF.

## Annotation of piRNAs

For the identification of piRNA clusters, small read RNA-seq was processed using the pipeline workflow described within the Pilfer (*Ray and Pandey, 2018*) toolkit to generate piRNA cluster BED files per sample. The sample piRNA clusters were then merged across samples to produce a final piRNA cluster set, which was then included in our integrated gene model GTF.

## Annotation of tRNAs, snRNAs, snoRNAs, and rRNAs

Additional annotations for tRNAs, snRNAs, snoRNAs, and rRNAs were mapped from RefSeq annotation onto our integrated gene model GTF.

## Sex-specific gene expression

To identify genes with sex-specific expression, we aligned Illumina mRNA-seq reads from three female and three male pupae using STAR (v2.5.0a) (*Dobin et al., 2013*). We counted and assigned reads to genes and transcripts using RSEM (v1.2.28) (*Li and Dewey, 2011*). We analyzed differential gene expression using DESeq2 (*Love et al., 2014*), filtering out genes shorter than 10 bases or with less than 10 counts across all six samples. We hierarchically clustered the differentially expressed genes by Euclidean distance using the R package pheatmap (v1.0.12) (*Kolde, 2024*). We used DEXSeq (v1.50.0) (*Anders et al., 2012*) to identify differential exon usage.

## Acknowledgements

We thank Stephany Valdés-Rodríguez, Leonora Olivos-Cisneros, Alejandra Hurtado-Giraldo, Sean McKenzie, and other members of the Kronauer laboratory for helping us obtain male ants and for valuable discussions; Jen Balacco and the Rockefeller University Reference Genome Resource Center for long read RNA sequencing; the Weill Cornell Genomics Resources Core Facility for Small RNA sequencing; Connie Zhao and the Rockefeller University Genomics Resource Center for rRNA-depletion sequencing and all other Illumina sequencing; Tom Kay, Matteo Rossi, and Yukina Chiba for valuable comments on an earlier version of the manuscript. This work was supported by a Gabrielle H Reem and Herbert J Kayden Early-Career Innovation Award and the National Institute of General Medical Sciences of the National Institutes of Health under award no. R35GM127007, both to DJCK. The content is solely the responsibility of the authors and does not necessarily represent the official views of the National Institutes of Health. This work was also supported by the Howard Hughes Medical Institute, where DJCK is an Investigator. This is Clonal Raider Ant Project paper number 36.

## Additional information

### Funding

| Funder | Grant reference number | Author |
|---|---|---|
| National Institute of General Medical Sciences | R35GM127007 | Daniel JC Kronauer |
| The Rockefeller University | Reem/Kayden Early-Career Innovation Award | Daniel JC Kronauer |
| Howard Hughes Medical Institute | Investigator Program | Daniel JC Kronauer |

The funders had no role in study design, data collection and interpretation, or the decision to submit the work for publication.

### Author contributions

Kip D Lacy, Conceptualization, Data curation, Software, Formal analysis, Validation, Investigation, Visualization, Methodology, Writing – original draft, Project administration, Writing – review and editing, Performed or supervised all research; Jina Lee, Data curation, Software, Formal analysis, Investigation, Visualization, Methodology, Writing – review and editing, Analyzed data and visualized results for gene-expression and exon usage analyses; Kathryn Rozen-Gagnon, Data curation, Software, Formal analysis, Investigation, Methodology, Improved the genome annotation; Wei Wang, Data curation, Software, Formal analysis, Investigation, Methodology, Improved the genome annotation; Thomas S Carroll, Data curation, Software, Formal analysis, Investigation, Methodology, Writing – review and editing, Improved the genome annotation; Daniel JC Kronauer, Conceptualization, Resources, Supervision, Funding acquisition, Validation, Writing – original draft, Project administration, Writing – review and editing

### Author ORCIDs
Kip D Lacy https://orcid.org/0000-0002-3149-8927
Daniel JC Kronauer https://orcid.org/0000-0002-4103-7729

Reviewer #1 (Public review): https://doi.org/10.7554/eLife.106913.3.sa1
Reviewer #3 (Public review): https://doi.org/10.7554/eLife.106913.3.sa2
Author response https://doi.org/10.7554/eLife.106913.3.sa3

## Additional files

### Supplementary files

Supplementary file 1. Primers for ploidy assessment via heterozygosity. 5.4 refers to the reference genome version (GCA_003672135.1).

Supplementary file 2. Male genotypes at ploidy markers.

Supplementary file 3. Metadata for all DNA whole-genome shotgun sequencing libraries included in this study. Question marks indicate male samples for which the clonal line is known, but the stock colony of origin was not recorded.

Supplementary file 4. Unique mutations and losses of heterozygosity the five diploid males without a large Chr4-linked loss of heterozygosity.

Supplementary file 5. *O. biroi* orthologs of genes located in the putative *L. fabarum* CSD 'regions'.

Supplementary file 6. *O. biroi* orthologs of genes located in the two *V. emeryi* CSD QTLs.

Supplementary file 7. All nucleotides in the CSD index peak that differ among clonal lines, including whether an amino acid differs.

Supplementary file 8. Improvements over previous genome annotations, with the new (RU) annotation featuring genes and transcripts not found in previous annotation versions.

Supplementary file 9. Full information on genome annotation improvements.

Supplementary file 10. Differential gene expression near the CSD index peak. The log 2-fold change in expression levels between male and female samples, p-values (Wald test), and adjusted p-values (Benjamini–Hochberg adjustment) for 17 genes in the vicinity of the CSD index peak. The p-values are 'NA' for LOC105283850 because one of the samples is an extreme outlier detected by Cook's distance.

Supplementary file 11. Differential exon usage near the CSD index peak. The log 2-fold change in expression levels between male and female samples, p-values (likelihood ratio test), and adjusted p-values (Benjamini–Hochberg adjustment) for each exon of 17 genes in the vicinity of the CSD index peak. The p-values are 'NA' for one of the exons of LOC105285603 because there is negligible expression of that exon (basemean = 0).

Supplementary file 12. Versions of software used for genomics analyses.

MDAR checklist

### Data availability

All DNA and RNA sequencing data are publicly available at the National Center for Biotechnology Information Sequence Read Archive under accession number PRJNA1075055. All other data are available in the article and Supporting Information. All code is available on GitHub (genome annotation: https://github.com/RockefellerUniversity/Obiroi_GeneModels_2025, copy archived at *Carroll and Wang, 2025*; differential gene expression and exon usage: https://github.com/jina-leemon/CSD_biroi_male-female-RNA-seq, copy archived at *Lee, 2025*; all other code and analyses: https://github.com/kipdlacy/SexDetermination_Obiroi, copy archived at *Lacy, 2025*).

The following dataset was generated:

| Author(s) | Year | Dataset title | Dataset URL | Database and Identifier |
| --- | --- | --- | --- | --- |
| Lacy KD, Kronauer DJC | 2024 | Genetic mapping of sex determination in the clonal raider ant | https://www.ncbi.nlm.nih.gov/bioproject/PRJNA1075055/ | NCBI BioProject, PRJNA1075055 |

The following previously published datasets were used:

| Author(s) | Year | Dataset title | Dataset URL | Database and Identifier |
|---|---|---|---|---|
| Trible W, Kronauer DJC | 2023 | Clonal raider ant Line A assemblies and resequencing | https://www.ncbi.nlm.nih.gov/bioproject/?term=PRJNA923657 | NCBI BioProject, PRJNA923657 |
| Lacy KD, Kronauer DJC | 2023 | Recombination and Heterozygosity Maintenance in Ooceraea biroi | https://www.ncbi.nlm.nih.gov/bioproject/?term=PRJNA947942 | NCBI BioProject, PRJNA947942 |

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
