## [Editor Report · eLife Assessment]

This study provides **valuable** insights into the evolutionary conservation of sex determination mechanisms in ants by identifying a candidate sex-determining region in a parthenogenetic species. It uses **solid**, well-executed genomic analyses based on differences in heterozygosity between females and diploid males. While the candidate locus awaits functional validation in this species, the study provides **convincing** support for the ancient origin of a non-coding locus implicated in sex determination.

---

## [Referee Report · Reviewer #1 (Public review)]

The authors have implemented several clarifications in the text and improved the connection between their findings and previous work. As stated in my initial review, I had no major criticisms of the previous version of the manuscript, and I continue to consider this a solid and well-written study. However, the revised manuscript still largely reiterates existing findings and does not offer novel conceptual or experimental advances. It supports previous conclusions suggesting a likely conserved sex determination locus in aculeate hymenopterans, but does so without functional validation (i.e., via experimental manipulation) of the candidate locus in O. biroi. I also wish to clarify that I did not intend to imply that functional assessments in the Pan et al. study were conducted in more than one focal species; my previous review explicitly states that the locus's functional role was validated in the Argentine ant.

---

## [Referee Report · Reviewer #3 (Public review)]

The authors have made considerable efforts to conduct functional analyses to the fullest extent possible in this study; however, it is understandable that meaningful results have not yet been obtained. In the revised version, they have appropriately framed their claims within the limits of the current data and have adjusted their statements as needed in response to the reviewers' comments.

---

## [Author Response]

The following is the authors’ response to the original reviews

**Reviewer #1 (Public review):**
This study investigates the sex determination mechanism in the clonal ant Ooceraea biroi, focusing on a candidate complementary sex determination (CSD) locus-one of the key mechanisms supporting haplodiploid sex determination in hymenopteran insects. Using whole genome sequencing, the authors analyze diploid females and the rarely occurring diploid males of O. biroi, identifying a 46 kb candidate region that is consistently heterozygous in females and predominantly homozygous in diploid males. This region shows elevated genetic diversity, as expected under balancing selection. The study also reports the presence of an lncRNA near this heterozygous region, which, though only distantly related in sequence, resembles the ANTSR lncRNA involved in female development in the Argentine ant, Linepithema humile (Pan et al. 2024). Together, these findings suggest a potentially conserved sex determination mechanism across ant species. However, while the analyses are well conducted and the paper is clearly written, the insights are largely incremental. The central conclusion - that the sex determination locus is conserved in ants - was already proposed and experimentally supported by Pan et al. (2024), who included O. biroi among the studied species and validated the locus's functional role in the Argentine ant. The present study thus largely reiterates existing findings without providing novel conceptual or experimental advances.

Although it is true that Pan *et al*., 2024 demonstrated (in Figure 4 of their paper) that the synteny of the region flanking *ANTSR* is conserved across aculeate Hymenoptera (including *O. biroi*), Reviewer 1’s claim that that paper provides experimental support for the hypothesis that the sex determination locus is conserved in ants is inaccurate. Pan *et al*., 2024 only performed experimental work in a single ant species (*Linepithema humile*) and merely compared reference genomes of multiple species to show synteny of the region, rather than functionally mapping or characterizing these regions.

Other comments:The mapping is based on a very small sample size: 19 females and 16 diploid males, and these all derive from a single clonal line. This implies a rather high probability for false-positive inference. In combination with the fact that only 11 out of the 16 genotyped males are actually homozygous at the candidate locus, I think a more careful interpretation regarding the role of the mapped region in sex determination would be appropriate. The main argument supporting the role of the candidate region in sex determination is based on the putative homology with the lncRNA involved in sex determination in the Argentine ant, but this argument was made in a previous study (as mentioned above).

Our main argument supporting the role of the candidate region in sex determination is not based on putative homology with the lncRNA in *L. humile.* Instead, our main argument comes from our genetic mapping (in Fig. 2), and the elevated nucleotide diversity within the identified region (Fig. 4). Additionally, we highlight that multiple genes within our mapped region are homologous to those in mapped sex determining regions in both *L. humile* and *Vollenhovia emeryi*, possibly including the lncRNA.

In response to the Reviewer’s assertion that the mapping is based on a small sample size from a single clonal line, we want to highlight that we used all diploid males available to us. Although the primary shortcoming of a small sample size is to increase the probability of a false negative, small sample sizes can also produce false positives. We used two approaches to explore the statistical robustness of our conclusions. First, we generated a null distribution by randomly shuffling sex labels within colonies and calculating the probability of observing our CSD index values by chance (shown in Fig. 2). Second, we directly tested the association between homozygosity and sex using Fisher’s Exact Test (shown in Supplementary Fig. S2). In both cases, the association of the candidate locus with sex was statistically significant after multiple-testing correction using the Benjamini-Hochberg False Discovery Rate. These approaches are clearly described in the “CSD Index Mapping” section of the Methods.

We also note that, because complementary sex determination loci are expected to evolve under balancing selection, our finding that the mapped region exhibits a peak of nucleotide diversity lends orthogonal support to the notion that the mapped locus is indeed a complementary sex determination locus.

The fourth paragraph of the results and the sixth paragraph of the discussion are devoted to explaining the possible reasons why only 11/16 genotyped males are homozygous in the mapped region. The revised manuscript will include an additional sentence (in what will be lines 384-388) in this paragraph that includes the possible explanation that this locus is, in fact, a false positive, while also emphasizing that we find this possibility to be unlikely given our multiple lines of evidence.

In response to Reviewer 1’s suggestion that we carefully interpret the role of the mapped region in sex determination, we highlight our careful wording choices, nearly always referring to the mapped locus as a “candidate sex determination locus” in the title and throughout the manuscript. For consistency, the revised manuscript version will change the second results subheading from “The *O. biroi* CSD locus is homologous to another ant sex determination locus but not to honeybee *csd*” to “*O. biroi*’s candidate CSD locus is homologous to another ant sex determination locus but not to honeybee *csd*,” and will add the word “candidate” in what will be line 320 at the beginning of the Discussion, and will change “putative” to “candidate” in what will be line 426 at the end of the Discussion.

In the abstract, it is stated that CSD loci have been mapped in honeybees and two ant species, but we know little about their evolutionary history. But CSD candidate loci were also mapped in a wasp with multi-locus CSD (study cited in the introduction). This wasp is also parthenogenetic via central fusion automixis and produces diploid males. This is a very similar situation to the present study and should be referenced and discussed accordingly, particularly since the authors make the interesting suggestion that their ant also has multi-locus CSD and neither the wasp nor the ant has tra homologs in the CSD candidate regions. Also, is there any homology to the CSD candidate regions in the wasp species and the studied ant?

In response to Reviewer 1’s suggestion that we reference the (Matthey-Doret et al. 2019) study in the context of diploid males being produced via losses of heterozygosity during asexual reproduction, the revised manuscript will include (in what will be lines 123-126) the highlighted portion of the following sentence: “Therefore, if *O. biroi* uses CSD, diploid males might result from losses of heterozygosity at sex determination loci (Fig. 1C), similar to what is thought to occur in other asexual Hymenoptera that produce diploid males (Rabeling and Kronauer 2012; Matthey-Doret et al. 2019).”

We note, however, that in their 2019 study, Matthey-Doret *et al*. did not directly test the hypothesis that diploid males result from losses of heterozygosity at CSD loci during asexual reproduction, because the diploid males they used for their mapping study came from inbred crosses in a sexual population of that species.

We address this further below, but we want to emphasize that we do not intend to argue that *O. biroi* has multiple CSD loci. Instead, we suggest that additional, undetected CSD loci is one possible explanation for the absence of diploid males from any clonal line other than clonal line A. In response to Reviewer 1’s suggestion that we reference the (Matthey-Doret et al. 2019) study in the context of multilocus CSD, the revised manuscript version will include the following additional sentence in the fifth paragraph of the discussion (in what will be lines 372-374): “Multi-locus CSD has been suggested to limit the extent of diploid male production in asexual species under some circumstances (Vorburger 2013; Matthey-Doret et al. 2019).”

Regarding Reviewer 2’s question about homology between the putative CSD loci from the (Matthey-Doret et al. 2019) study and *O. biroi*, we note that there is no homology. The revised manuscript version will have an additional Supplementary Table (which will be the new Supplementary Table S3) that will report the results of this homology search. The revised manuscript will also include the following additional sentence in the Results, in what will be lines 172-174: “We found no homology between the genes within the *O. biroi* CSD index peak and any of the genes within the putative *L. fabarum* CSD loci (Supplementary Table S3).”

The authors used different clonal lines of O. biroi to investigate whether heterozygosity at the mapped CSD locus is required for female development in all clonal lines of O. biroi (L187-196). However, given the described parthenogenesis mechanism in this species conserves heterozygosity, additional females that are heterozygous are not very informative here. Indeed, one would need diploid males in these other clonal lines as well (but such males have not yet been found) to make any inference regarding this locus in other lines.

We agree that a full mapping study including diploid males from all clonal lines would be preferable, but as stated earlier in that same paragraph, we have only found diploid males from clonal line A. We stand behind our modest claim that “Females from all six clonal lines were heterozygous at the CSD index peak, consistent with its putative role as a CSD locus in all *O. biroi.”* In the revised manuscript version, this sentence (in what will be lines 199-201) will be changed slightly in response to a reviewer comment below: “All females from all six clonal lines (including 26 diploid females from clonal line B) were heterozygous at the CSD index peak, consistent with its putative role as a CSD locus in all *O. biroi*.”

**Reviewer #2 (Public review):**
The manuscript by Lacy et al. is well written, with a clear and compelling introduction that effectively conveys the significance of the study. The methods are appropriate and well-executed, and the results, both in the main text and supplementary materials, are presented in a clear and detailed manner. The authors interpret their findings with appropriate caution.This work makes a valuable contribution to our understanding of the evolution of complementary sex determination (CSD) in ants. In particular, it provides important evidence for the ancient origin of a non-coding locus implicated in sex determination, and shows that, remarkably, this sex locus is conserved even in an ant species with a non-canonical reproductive system that typically does not produce males. I found this to be an excellent and well-rounded study, carefully analyzed and well contextualized.That said, I do have a few minor comments, primarily concerning the discussion of the potential 'ghost' CSD locus. While the authors acknowledge (line 367) that they currently have no data to distinguish among the alternative hypotheses, I found the evidence for an additional CSD locus presented in the results (lines 261-302) somewhat limited and at times a bit difficult to follow. I wonder whether further clarification or supporting evidence could already be extracted from the existing data. Specifically:

We agree with Reviewer 2 that the evidence for a second CSD locus is limited. In fact, we do not intend to advocate for there being a second locus, but we suggest that a second CSD locus is one possible explanation for the absence of diploid males outside of clonal line A. In our initial version, we intentionally conveyed this ambiguity by titling this section “*O. biroi* may have one or multiple sex determination loci.” However, we now see that this leads to undue emphasis on the possibility of a second locus. In the revised manuscript, we will split this into two separate sections: “Diploid male production differs across *O. biroi* clonal lines” and “*O. biroi* lacks a *tra*-containing CSD locus.”

(1) Line 268: I doubt the relevance of comparing the proportion of diploid males among all males between lines A and B to infer the presence of additional CSD loci. Since the mechanisms producing these two types of males differ, it might be more appropriate to compare the proportion of diploid males among all diploid offspring. This ratio has been used in previous studies on CSD in Hymenoptera to estimate the number of sex loci (see, for example, Cook 1993, de Boer et al. 2008, 2012, Ma et al. 2013, and Chen et al., 2021). The exact method might not be applicable to clonal raider ants, but I think comparing the percentage of diploid males among the total number of (diploid) offspring produced between the two lineages might be a better argument for a difference in CSD loci number.

We want to re-emphasize here that we do not wish to advocate for there being two CSD loci in *O. biroi*. Rather, we want to explain that this is one possible explanation for the apparent absence of diploid males outside of clonal line A. We hope that the modifications to the manuscript described in the previous response help to clarify this.

Reviewer 2 is correct that comparing the number of diploid males to diploid females does not apply to clonal raider ants. This is because males are vanishingly rare among the vast numbers of females produced. We do not count how many females are produced in laboratory stock colonies, and males are sampled opportunistically. Therefore, we cannot report exact numbers. However, we will add the highlighted portion of the following sentence (in what will be lines 268-270) to the revised manuscript: “Despite the fact that we maintain more colonies of clonal line B than of clonal line A in the lab, all the diploid males we detected came from clonal line A.”

(2) If line B indeed carries an additional CSD locus, one would expect that some females could be homozygous at the ANTSR locus but still viable, being heterozygous only at the other locus. Do the authors detect any females in line B that are homozygous at the ANTSR locus? If so, this would support the existence of an additional, functionally independent CSD locus.

We thank the reviewer for this suggestion, and again we emphasize that we do not want to argue in favor of multiple CSD loci. We just want to introduce it as one possible explanation for the absence of diploid males outside of clonal line A.

The 26 sequenced diploid females from clonal line B are all heterozygous at the mapped locus, and the revised manuscript will clarify this in what will be lines 199-201. Previously, only six of those diploid females were included in Supplementary Table S2, and that will be modified accordingly.

(3) Line 281: The description of the two tra-containing CSD loci as "conserved" between Vollenhovia and the honey bee may be misleading. It suggests shared ancestry, whereas the honey bee csd gene is known to have arisen via a relatively recent gene duplication from fem/tra (10.1038/nature07052). It would be more accurate to refer to this similarity as a case of convergent evolution rather than conservation.

In the sentence that Reviewer 2 refers to, we are representing the assertion made in the (Miyakawa and Mikheyev 2015) paper in which, regarding their mapping of a candidate CSD locus that contains two linked *tra* homologs, they write in the abstract: “these data support the prediction that the same CSD mechanism has indeed been conserved for over 100 million years.” In that same paper, Miyakawa and Mikheyev write in the discussion section: “As ants and bees diverged more than 100 million years ago, sex determination in honey bees and V. emeryi is probably homologous and has been conserved for at least this long.”

As noted by Reviewer 2, this appears to conflict with a previously advanced hypothesis: that because *fem* and *csd* were found in *Apis mellifera*, *Apis cerana*, and *Apis dorsata*, but only *fem* was found in *Mellipona compressipes*, *Bombus terrestris*, and *Nasonia vitripennis*, that the *csd* gene evolved after the honeybee (*Apis*) lineage diverged from other bees (Hasselmann et al. 2008). However, it remains possible that the *csd* gene evolved after ants and bees diverged from *N. vitripennis*, but before the divergence of ants and bees, and then was subsequently lost in *B. terrestris* and *M. compressipes*. This view was previously put forward based on bioinformatic identification of putative orthologs of *csd* and *fem* in bumblebees and in ants [(Schmieder et al. 2012), see also (Privman et al. 2013)]. However, subsequent work disagreed and argued that the duplications of *tra* found in ants and in bumblebees represented convergent evolution rather than homology (Koch et al. 2014). Distinguishing between these possibilities will be aided by additional sex determination locus mapping studies and functional dissection of the underlying molecular mechanisms in diverse Aculeata.

Distinguishing between these competing hypotheses is beyond the scope of our paper, but the revised manuscript will include additional text to incorporate some of this nuance. We will include these modified lines below (in what will be lines 287-295), with the additions highlighted:

“A second QTL region identified in *V. emeryi* (*V.emeryiCsdQTL1*) contains two closely linked *tra* homologs, similar to the closely linked honeybee *tra* homologs, *csd* and *fem* (Miyakawa and Mikheyev 2015). This, along with the discovery of duplicated *tra* homologs that undergo concerted evolution in bumblebees and ants (Schmieder et al. 2012; Privman et al. 2013) has led to the hypothesis that the function of *tra* homologs as CSD loci is conserved with the *csd*-containing region of honeybees (Schmieder et al. 2012; Miyakawa and Mikheyev 2015). However, other work has suggested that *tra* duplications occurred independently in honeybees, bumblebees, and ants (Hasselmann et al. 2008; Koch et al. 2014), and it remains to be demonstrated that either of these *tra* homologs acts as a primary CSD signal in *V. emeryi*.”

(4) Finally, since the authors successfully identified multiple alleles of the first CSD locus using previously sequenced haploid males, I wonder whether they also observed comparable allelic diversity at the candidate second CSD locus. This would provide useful supporting evidence for its functional relevance.Overall, these are relatively minor points in the context of a strong manuscript, but I believe addressing them would improve the clarity and robustness of the authors' conclusions.

As is already addressed in the final paragraph of the results and in Supplementary Fig. S4, there is no peak of nucleotide diversity in any of the regions homologous to *V.emeryiQTL1*, which is the *tra*-containing candidate sex determination locus (Miyakawa and Mikheyev 2015). In the revised manuscript, the relevant lines will be 307-310. We want to restate that we do not propose that there is a second candidate CSD locus in *O. biroi*, but we simply raise the possibility that multi-locus CSD *might* explain the absence of diploid males from clonal lines other than clonal line A (as one of several alternative possibilities).

**Reviewer #3 (Public review):**
Summary:The sex determination mechanism governed by the complementary sex determination (CSD) locus is one of the mechanisms that support the haplodiploid sex determination system evolved in hymenopteran insects. While many ant species are believed to possess a CSD locus, it has only been specifically identified in two species. The authors analyzed diploid females and the rarely occurring diploid males of the clonal ant Ooceraea biroi and identified a 46 kb CSD candidate region that is consistently heterozygous in females and predominantly homozygous in males. This region was found to be homologous to the CSD locus reported in distantly related ants. In the Argentine ant, Linepithema humile, the CSD locus overlaps with an lncRNA (ANTSR) that is essential for female development and is associated with the heterozygous region (Pan et al. 2024). Similarly, an lncRNA is encoded near the heterozygous region within the CSD candidate region of O. biroi. Although this lncRNA shares low sequence similarity with ANTSR, its potential functional involvement in sex determination is suggested. Based on these findings, the authors propose that the heterozygous region and the adjacent lncRNA in O. biroi may trigger female development via a mechanism similar to that of L. humile. They further suggest that the molecular mechanisms of sex determination involving the CSD locus in ants have been highly conserved for approximately 112 million years. This study is one of the few to identify a CSD candidate region in ants and is particularly noteworthy as the first to do so in a parthenogenetic species.Strengths:(1) The CSD candidate region was found to be homologous to the CSD locus reported in distantly related ant species, enhancing the significance of the findings.(2) Identifying the CSD candidate region in a parthenogenetic species like O. biroi is a notable achievement and adds novelty to the research.Weaknesses(1) Functional validation of the lncRNA's role is lacking, and further investigation through knockout or knockdown experiments is necessary to confirm its involvement in sex determination.

See response below.

(2) The claim that the lncRNA is essential for female development appears to reiterate findings already proposed by Pan et al. (2024), which may reduce the novelty of the study.

We do not claim that the lncRNA is essential for female development in *O. biroi*, but simply mention the possibility that, as in *L. humile*, it is somehow involved in sex determination. We do not have any functional evidence for this, so this is purely based on its genomic position immediately adjacent to our mapped candidate region. We agree with the reviewer that the study by Pan et al. (2024) decreases the novelty of our findings. Another way of looking at this is that our study supports and bolsters previous findings by partially replicating the results in a different species.

**Recommendations for the authors:**

**Reviewer #1 (Recommendations for the authors):**
L307-308 should state homozygous for either allele in THE MAJORITY of diploid males.

This will be fixed in the revised manuscript, in what will be line 321.

**Reviewer #3 (Recommendations for the authors):**
The association between heterozygosity in the CSD candidate region and female development in O. biroi, along with the high sequence homology of this region to CSD loci identified in two distantly related ant species, is not sufficient to fully address the evolution of the CSD locus and the mechanisms of sex determination.Given that functional genetic tools, such as genome editing, have already been established in O. biroi, I strongly recommend that the authors investigate the role of the lncRNA through knockout or knockdown experiments and assess its impact on the sex-specific splicing pattern of the downstream tra gene.

Although knockout experiments of the lncRNA would be illuminating, the primary signal of complementary sex determination is heterozygosity. As is clearly stated in our manuscript and that of (Pan et al. 2024), it does not appear to be heterozygosity within the lncRNA that induces female development, but rather heterozygosity in non-transcribed regions linked to the lncRNA. Therefore, future mechanistic studies of sex determination in *O. biroi*, *L. humile*, and other ants should explore how homozygosity or heterozygosity of this region impacts the sex determination cascade, rather than focusing (exclusively) on the lncRNA.

With this in mind, we developed three sets of guide RNAs that cut only one allele within the mapped CSD locus, with the goal of producing deletions within the highly variable region within the mapped locus. This would lead to functional hemizygosity or homozygosity within this region, depending on how the cuts were repaired. We also developed several sets of PCR primers to assess the heterozygosity of the resultant animals. After injecting 1,162 eggs over several weeks and genotyping the hundreds of resultant animals with PCR, we confirmed that we could induce hemizygosity or homozygosity within this region, at least in ~1/20 of the injected embryos. Although it is possible to assess the sex-specificity of the splice isoform of *tra* as a proxy for sex determination phenotypes (as done by (Pan et al. 2024)), the ideal experiment would assess male phenotypic development at the pupal stage. Therefore, over several more weeks, we injected hundreds more eggs with these reagents and reared the injected embryos to the pupal stage. However, substantial mortality was observed, with only 12 injected eggs developing to the pupal stage. All of these were female, and none of them had been successfully mutated.

In conclusion, we agree with the reviewer that functional experiments would be useful, and we made extensive attempts to conduct such experiments. However, these experiments turned out to be extremely challenging with the currently available protocols. Ultimately, we therefore decided to abandon these attempts.

We opted not to include these experiments in the paper itself because we cannot meaningfully interpret their results. However, we are pleased that, in this response letter, we can include a brief description for readers interested in attempting similar experiments.

Since O. biroi reproduces parthenogenetically and most offspring develop into females, observing a shift from female- to male-specific splicing of tra upon early embryonic knockout of the lncRNA would provide much stronger evidence that this lncRNA is essential for female development. Without such functional validation, the authors' claim (lines 36-38) seems to reiterate findings already proposed by Pan et al. (2024) and, as such, lacks sufficient novelty.

We have responded to the issue of “lack of novelty” above. But again, the actual CSD locus in both *O. biroi* and *L. humile* appears to be distinct from (but genetically linked to) the lncRNA, and we have no experimental evidence that the putative lncRNA in *O. biroi* is involved in sex determination at all. Because of this, and given the experimental challenges described above, we do not currently intend to pursue functional studies of the lncRNA.

References

Hasselmann M, Gempe T, Schiøtt M, Nunes-Silva CG, Otte M, Beye M. 2008. Evidence for the evolutionary nascence of a novel sex determination pathway in honeybees. Nature 454:519–522.

Koch V, Nissen I, Schmitt BD, Beye M. 2014. Independent Evolutionary Origin of fem Paralogous Genes and Complementary Sex Determination in Hymenopteran Insects. PLOS ONE 9:e91883.

Matthey-Doret C, van der Kooi CJ, Jeffries DL, Bast J, Dennis AB, Vorburger C, Schwander T. 2019. Mapping of multiple complementary sex determination loci in a parasitoid wasp. Genome Biology and Evolution 11:2954–2962.

Miyakawa MO, Mikheyev AS. 2015. QTL mapping of sex determination loci supports an ancient pathway in ants and honey bees. PLOS Genetics 11:e1005656.

Pan Q, Darras H, Keller L. 2024. LncRNA gene ANTSR coordinates complementary sex determination in the Argentine ant. Science Advances 10:eadp1532.

Privman E, Wurm Y, Keller L. 2013. Duplication and concerted evolution in a master sex determiner under balancing selection. Proceedings of the Royal Society B: Biological Sciences 280:20122968.

Rabeling C, Kronauer DJC. 2012. Thelytokous parthenogenesis in eusocial Hymenoptera. Annual Review of Entomology 58:273–292.

Schmieder S, Colinet D, Poirié M. 2012. Tracing back the nascence of a new sex-determination pathway to the ancestor of bees and ants. Nature Communications 3:1–7.

Vorburger C. 2013. Thelytoky and Sex Determination in the Hymenoptera: Mutual Constraints. Sexual Development 8:50–58.